REGISTERED REPORT PROTOCOL

# Characterising the influence of cerebellum on the neuroplastic modulation of intracortical motor circuits

**George M. Opie** *, **John G. Semmler**

Discipline of Physiology, Adelaide Medical School, The University of Adelaide, Adelaide, Australia

* george.opie@adelaide.edu.au

## Abstract

The cerebellum (CB) has extensive connections with both cortical and subcortical areas of the brain, and is known to strongly influence function in areas it projects to. In particular, research using non-invasive brain stimulation (NIBS) has shown that CB projections to primary motor cortex (M1) are likely important for facilitating the learning of new motor skills, and that this process may involve modulation of late indirect (I) wave inputs in M1. However, the nature of this relationship remains unclear, particularly in regards to how CB influences the contribution of the I-wave circuits to neuroplastic changes in M1. Within the proposed research, we will therefore investigate how CB effects neuroplasticity of the I-wave generating circuits. This will be achieved by downregulating CB excitability while concurrently applying a neuroplastic intervention that specifically targets the I-wave circuitry. The outcomes of this study will provide valuable neurophysiological insight into key aspects of the motor network, and may inform the development of optimized interventions for modifying motor learning in a targeted way.

## Introduction

The ability to modify patterns of motor behaviour in response to sensory feedback represents a fundamental component of effective motor control. This process underpins our capacity to learn new types of motor skills, and to improve their performance with practice. While this error-based motor adaptation is a complex process involving a distributed brain network, extensive literature has shown that the cerebellum (CB) plays a critical role (for review, see; [1]). This structure is thought to facilitate generation and ongoing modification of internal models of neural activation that determine effector dynamics. These internal models are constantly updated based on comparisons between predicted and actual sensory feedback, allowing improved task performance with practice. As an extension of this process, communication between CB and primary motor cortex (M1) is crucial [2, 3], and may facilitate retention of the generated internal model [4]. However, the neurophysiological processes underpinning this communication remain unclear, largely due to the difficulty of assessing the associated pathways in human participants.

**Data Availability Statement:** Upon completion of the study, all data will be made fully available via

hosting on the open science framework repository (https://www.cos.io/osf).

**Funding:** The author(s) received no specific funding for this work.

**Competing interests:** The authors have declared that no competing interests exist.

Despite this, non-invasive brain stimulation techniques (NIBS) such as transcranial magnetic stimulation (TMS) have provided some information on CB-M1 communication. In particular, inhibitory interactions between CB and M1 have been demonstrated using a paradigm called CB-brain inhibition (CBI). This involves applying a TMS pulse over the CB at specific intervals (5–7 ms) prior to a second stimulus over contralateral M1, producing a motor evoked potential (MEP) that is reduced in amplitude relative to an MEP produced by M1 stimulation alone [5–7]. CBI is thought to involve activation of purkinje cells in CB cortex, leading to inhibition of the dentate nucleus and consequent disfacilitation of M1 via projections through the motor thalamus (for review, see; [8]). Activity of this pathway is known to be modified during the learning of adaptation tasks that rely heavily on input from the CB [9–11], with larger changes in CBI predicting better performance [11]. Furthermore, manipulating activity in the CB-M1 pathway can influence neuroplasticity in M1 [12–14]. Consequently, CB-M1 connections are important for the acquisition of new motor skills, and CB-dependent changes in M1 neuroplasticity are one way by which CB may influence motor output.

While this literature demonstrates the capacity of CB to influence M1 plasticity in a functionally relevant way, it remains unclear how this influence is mediated. In particular, the circuits within M1 that are targeted by CB are unknown. Given that previous research using TMS has shown that the activity of specific intracortical motor circuits relates to the acquisition of different motor skills [15], identification of the M1 circuitry that is affected by CB projections may allow the targeted modification of skill acquisition. Interestingly, growing evidence suggests that late indirect (I) wave inputs on to corticospinal neurons, which represent important predictors of neuroplasticity and motor learning [15–17], may be specifically modified by changes in CB excitability. For example, application of transcranial direct current stimulation (tDCS; a NIBS paradigm that induces neuroplastic changes in brain excitability) over CB specifically modulates paired-pulse TMS measures of late I-wave excitability [18]. In addition, the effects of CB tDCS on single-pulse TMS measures of M1 excitability are only apparent when stimulation is applied with an anterior-posterior current, which specifically activates late I-wave circuits [19]. Also, changes in late I-wave circuits following motor training were observed following a CB-dependent motor task, but were absent following a task with minimal CB involvement [15].

Based on this previous literature, it appears likely that CB projections to M1 influence activity within the late I-wave circuitry. However, the nature of this influence, particularly in relation to the plasticity of these circuits, remains unclear. The aim of this exploratory study is therefore to assess how changes in CB activity influence the excitability and plasticity of I-wave generating circuits in M1. To achieve this, CB excitability will be downregulated using cathodal tDCS, whereas plasticity targeting the early and late I-wave circuits will be concurrently induced using I-wave periodicity TMS (iTMS;[20, 21]).

## Methods

### Sample size and participants

While the effects of CB tDCS on iTMS have not been previously investigated, the study by Ates and colleagues [18] investigated the influence of CB tDCS on the excitability of the I-wave generating circuits. Consequently, sample size calculations based on this study are sufficient to demonstrate the effects of activation within the pathway of interest (i.e., cerebellar projections to I-wave circuits of M1). Examination of the findings reported by Ates and colleagues revealed that changes in short-interval intracortical facilitation (SICF; paired-pulse TMS protocol indexing I-wave excitability; [22, 23]) due to CB tDCS had an effect size of 0.67. Based on

the results of an *a priori* power analysis utilising this effect size, with $\alpha = 0.05$ and $1\text{-}\beta = 0.9$, we will therefore recruit 21 individuals (half female) to participate in the proposed experiment.

All participants will be recruited via advertisements placed on notice boards within the University of Adelaide, in addition to on social media platforms. Exclusion criteria will include a history of psychiatric or neurological disease, current use of medications that effect the central nervous system, or left handedness. Suitability to receive TMS will be assessed using a standard screening questionnaire [24]. The experiment will be conducted in accordance with the Declaration of Helsinki, and has been approved by the University of Adelaide Human Research Ethics Committee (approval H-2019-252). Written, informed consent will be provided prior to participation. Upon completion of the study, all data will be made fully available via hosting on the open science framework repository (https://www.cos.io/osf).

## Experimental arrangement

All participants will attend the laboratory for three separate sessions, with a washout period of at least 1 week between sessions. While the experimental protocol applied within each session will be the same, the ISI used for iTMS will vary between sessions (see below & Fig 1). Furthermore, the order in which each iTMS interval is applied will be randomised between participants. As diurnal variations in cortisol can influence the neuroplastic response to TMS, all plasticity interventions will be applied after 11 am and at approximately the same time of day within each participant.

During each experimental session, the participant will be seated in a comfortable chair with their hand resting on a table in front of them. Surface electromyography (EMG) will be recorded from the first dorsal interosseous (FDI) of the right hand using two Ag-AgCl electrodes arranged in a belly-tendon montage on the skin above the muscle. A third electrode attached above the styloid process of the right ulnar will ground the electrodes. EMG signals will be amplified (300x) and filtered (band-pass 20Hz– 1 kHz) using a CED 1902 signal conditioner (Cambridge Electronic Design, Cambridge, UK) before being digitized at 2 kHz using CED 1401 analogue-to-digital converter and stored on a PC for offline analysis. Signal noise associated with mains power (within the 50 Hz frequency band) will also be removed using a Humbug mains noise eliminator (Quest Scientific, North Vancouver, Canada). To facilitate muscle relaxation when required, real-time EMG signals will be displayed on an oscilloscope placed in front of the participant.

## Experimental procedures

**Transcranial magnetic stimulation (TMS).** A figure-of-8 coil connected to two Magstim 200$^2$ magnetic stimulators (Magstim, Dyfed, UK) via a BiStim unit will be used to apply TMS to the left M1. The coil will be held tangentially to the scalp, at an angle of 45˚ to the sagittal

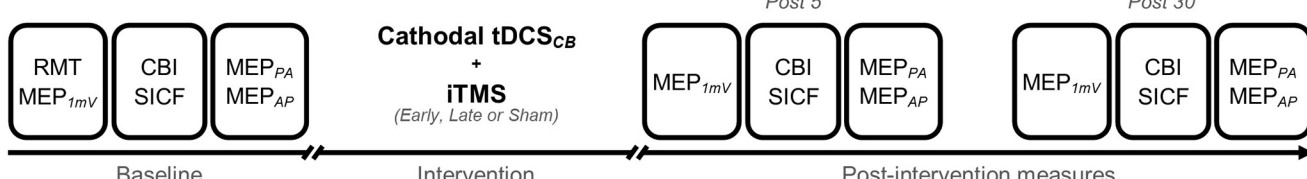

**Fig 1. Experimental protocol.** RMT, resting motor threshold; MEP$_{1mV}$, standard MEP of ~ 1 mV at baseline; CBI, cerebellar-brain inhibition; SICF, short-interval intracortical facilitation; MEP$_{PA}$, standard MEP of ~ 0.5 mV at baseline with PA orientation; MEP$_{AP}$, standard MEP of ~ 0.5 mV at baseline with AP orientation; tDCS$_{CB}$, transcranial direct current stimulation applied to the cerebellum; iTMS, I-wave periodicity repetitive transcranial magnetic stimulation.

plane, with the handle pointing backwards and laterally, inducing a posterior-to-anterior (PA) current within the brain. The location producing the largest and most consistent motor evoked potential (MEP) within the relaxed FDI muscle of the right hand will be identified and marked on the scalp for reference; this target location will be closely monitored throughout the experiment. All pre- and post-intervention TMS will be applied at a rate of 0.2 Hz, with a 10% jitter between trials in order to avoid anticipation of the stimulus.

Resting motor threshold (RMT) will be defined as the stimulus intensity producing an MEP amplitude $\geq 50$ µV in at least 5 out of 10 trials during relaxation of the right FDI. RMT will be assessed at the beginning of each experimental session and expressed as a percentage of maximum stimulator output (%MSO). Following assessment of RMT, the stimulus intensity producing a standard MEP amplitude of approximately 1 mV ($MEP_{1mV}$), when averaged over 20 trials, will be identified. The same intensity will then be applied 5 mins and 30 mins following the intervention in order to assess changes in corticospinal excitability.

**I-wave excitability.** As assessing the influence of CB modulation on I-wave excitability is the main aim of this project, changes in SICF will be the primary outcome measure. This paired-pulse TMS protocol produces MEP facilitation when conditioning and test stimuli are separated by discrete ISIs that correspond to I-wave latencies recorded from the epidural space [22]. SICF will utilise a conditioning stimulus set at 90% RMT, a test stimulus set at $MEP_{1mV}$ and two ISIs of 1.5 and 4.5 ms, which correspond to the early and late MEP peaks apparent in a complete SICF curve [22, 25, 26]. Measurements of SICF will include 12 trials for each condition, at each time point.

As a secondary measure of I-wave function, TMS will be applied with different stimulus directions, which alters the interneuronal circuits contributing to the generated MEP [16, 27, 28]. When TMS is applied with a conventional (PA) current direction, the resulting MEP is thought to arise from preferential activation of early I1-waves. In contrast, when the induced current is directed from anterior-to-posterior (AP; coil handle held 180° to the PA orientation), the resulting MEP is thought to arise from preferential activation of later (I2-3) I-waves. The stimulus intensity producing an MEP of approximately 0.5 mV will be assessed for both PA ($MEP_{PA}$) and AP ($MEP_{AP}$) orientations at baseline. The same intensities will then be reapplied 5 mins and 30 mins after the intervention, with 20 trials applied at each time point. While the I-wave specificity of these measures is generally suggested to rely on concurrent activation of the target muscle [27], post-intervention muscle activation is also known to strongly influence neuroplasticity induction [29–31]. As the current study is primarily concerned with plasticity induction, these measures will therefore be applied in a resting muscle in order to minimize confounding effects of voluntary contraction. Given the likely independence of the intracortical circuits activated with different currents, these measures will still provide useful physiological insight.

**Cerebellar-brain inhibition (CBI).** The strength of CB's inhibitory influence on M1 will be assessed using CBI, a stimulation protocol involving a conditioning stimulus applied to CB 5 ms prior to a test stimulus applied to M1 [5]. In accordance with previous literature, CB stimulation will be applied using a double cone coil, with the center of the coil located 3 cm lateral and 1 cm inferior to the inion, along the line joining the inion and the external auditory meatus of the right ear. The coil current will be directed downward, resulting in an upward induced current. The intensity of CB stimulation will be set at 60% MSO [32], whereas M1 stimulation will be set at $MEP_{1mV}$. The dual coil configuration of this measurement will mean that each coil will be directly connected to an individual Magstim $200^2$ stimulator. As removing the BiStim unit will result in an increase in stimulus strength, the $MEP_{1mV}$ intensity will be checked prior to baseline CBI measures, and adjusted when required. Because antidromic activation of corticospinal neurons may confound measures of CBI [33], we will ensure that the

CB conditioning stimulus is at least 5% MSO below the active motor threshold for the corticospinal tract [34]. Measures of CBI will be assessed at baseline, 5 mins and 30 mins post-intervention, with 15 trials recorded for each condition at each time point.

**I-wave periodicity repetitive TMS (iTMS).** In accordance with previous literature [21, 35], iTMS will consist of 180 pairs of stimuli applied every 5 s, resulting in a total intervention time of 15 mins. The same intensity will be used for both stimuli, adjusted to produce a response of ~ 1mV when applied as a pair. These stimuli will be applied using ISI's of 1.5 (iTMS$_{Early}$) and 4.5 ms (iTMS$_{Late}$) in separate sessions. These parameters produce robust potentiation of MEP amplitude [20, 21, 35, 36]. A sham stimulation condition (iTMS$_{Sham}$) that is not expected to modulate corticospinal excitability will also be applied in a third session. Within this condition, we will stimulate intervals that correspond to the transition between the peaks and troughs of facilitation that are observed within a complete SICF curve, as these are not expected to induce any changes in excitability. This will include equal repetitions of 1.8, 2.3, 3.3, 3.8 and 4.7 ms ISI's, applied randomly and with an inter-trial jitter of 10%.

**Cerebellar transcranial direct current stimulation (tDCS$_{CB}$).** A Soterix Medical 1 x 1 DC stimulator (Soterix Medical, New York, NY) will be used to apply tDCS to CB. Current will be applied through saline-soaked sponge electrodes (EASYpads, 5 x 7 cm), with the cathode positioned over the same location used for CB TMS (i.e., 3 cm lateral and 1 cm inferior to inion, contralateral to M1 TMS) and anode positioned on the skin above the right Buccinator muscle [4, 13, 37]. Stimulation will be applied at an intensity of 2 mA for 15 mins [4, 13, 37], coincident with the application of iTMS to M1. Onset and offset of stimulation will be ramped over a 30 s period prior to and following iTMS application.

## Data analysis

Analysis of EMG data will be completed manually via visual inspection of offline recordings. For measures in resting muscle, any trials with EMG activity exceeding 25 μV in the 100 ms prior to stimulus application will be excluded from analysis. All MEPs will be measured peak-to-peak and expressed in mV. Measures of CBI will be quantified by expressing the amplitude of individual trials produced by paired-pulse stimulation as a percentage of the mean response produced by single-pulse stimulation within the same block. For baseline measures of SICF, individual trials produced by paired-pulse stimulation will be expressed as a percentage of the mean response produced by single-pulse stimulation within the same block. However, for post-intervention responses, previous work has suggested that increased facilitation following iTMS correlates with the increased response to single pulse stimulation, and that this relationship cancels the effects of iTMS on SICF if the post-intervention single-pulse MEPs are used to normalise post-intervention SICF values [21]. Consequently, prior to normalization of post-intervention SICF values, we will use linear regression analysis to assess the relationship between post-intervention single and paired-pulse responses. If the relationship is significant, individual post-intervention SICF trials will be expressed relative to the mean pre-intervention single-pulse MEP [21]. However, if the relationship is not significant, post-intervention SICF measures will be expressed relative to the single-pulse response recorded in the same block. For all TMS measures, effects of the intervention will be quantified by expressing the post-intervention responses (normalised to the relevant single-pulse response for CBI and SICF) as a percentage of the pre-intervention responses.

## Statistical analysis

Kolmogorov-Smirnov tests will be applied to assess data distribution prior to statistical analysis, with log transformation applied when deviations from normal are indicated. Given that all

data of interest will involve repeated-measures, linear mixed-model analysis with repeated-measures ($LMM_{RM}$) will be used to perform all statistical comparisons. Each model will include single trial data, and previously reported methods [38] will be used to justify inclusion of random participant intercepts and/or slopes, and to optimize and assess model fit. To ensure measures are comparable between sessions, effects of iTMS session ($iTMS_{1.5}$, $iTMS_{4.5}$ & $iTMS_{Sham}$) on baseline measures of $MEP_{1mv}$, $MEP_{PA}$, $MEP_{AP}$ and CBI, in addition to responses recorded at the start of the iTMS intervention, will be investigated using one-factor $LMM_{RM}$, with each measurement investigated in a separate model. Furthermore, effects of iTMS session and ISI (1.5 & 4.5 ms) on baseline SICF will be assessed using two-factor $LMM_{RM}$.

Changes in excitability during the intervention will be assessed by comparing values averaged over the first, middle and last 12 stimuli of the iTMS block between iTMS sessions. Changes in corticospinal excitability following the intervention will be investigated by assessing the effects of iTMS session and time (Post 5, Post 30) on baseline-normalised $MEP_{1mV}$ values. Changes in coil-orientation dependent measures of I-wave excitability following the intervention will be investigated by assessing effects of iTMS session, time and coil orientation ($MEP_{PA}$, $MEP_{AP}$) on baseline-normalised values. Changes in SICF measures of I-wave excitability following the intervention will be investigated by assessing effects of iTMS session, time and ISI on baseline-normalised values. Changes in CBI following the intervention will be investigated by assessing the effects of iTMS session and time on baseline-normalised CBI values. For all models, investigation of main effects and interactions will be performed using custom contrasts with Bonferroni correction, and significance will be set at $P < 0.05$. Data for all models will be presented as the estimated marginal means, whereas pairwise comparisons will be presented as the estimated mean difference (EMD) and 95% confidence interval (95%CI) for the estimate, providing a non-standardised measure of effect size.

Regression analyses will be used to investigate interactions between variables. Specifically, changes in CBI due to the intervention will be regressed against changes in measures of corticospinal and intracortical function in order to assess if alterations within the CB-M1 pathway contribute to plasticity effects. In addition, changes in intracortical function due to the intervention will be regressed against changes in corticospinal function in an attempt to identify if generalised changes in excitability are driven by alterations within specific circuits.

## Author Contributions

**Conceptualization:** George M. Opie, John G. Semmler.

**Funding acquisition:** George M. Opie.

**Methodology:** George M. Opie.

**Project administration:** George M. Opie.

**Resources:** George M. Opie.

**Supervision:** John G. Semmler.

**Writing – original draft:** George M. Opie.

**Writing – review & editing:** George M. Opie, John G. Semmler.

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
