## [Decision Letter · Decision Letter 0]

22 May 2020

PONE-D-20-08985

Characterising the influence of cerebellum on the neuroplastic modulation of intracortical motor circuits.

PLOS ONE

Dear Dr Opie,

Thank you for submitting your manuscript to PLOS ONE. After careful consideration, we feel that it has merit but does not fully meet PLOS ONE’s publication criteria as it currently stands. Therefore, we invite you to submit a revised version of the manuscript that addresses the points raised during the review process.

Both reviewers found that the proposed study is interesting and worth pursing. However, several aspects need to be clarified including (but not limited to) the hypotheses, nature of sham stimulation and sample size calculations. Please refer to details in the reviewers' report.

We would appreciate receiving your revised manuscript by Jul 06 2020 11:59PM. To enhance the reproducibility of your results, we recommend that if applicable you deposit your laboratory protocols in protocols.io, where a protocol can be assigned its own identifier (DOI) such that it can be cited independently in the future. For instructions see: http://journals.plos.org/plosone/s/submission-guidelines#loc-laboratory-protocols

We look forward to receiving your revised manuscript.

Kind regards,

Robert Chen

Academic Editor

PLOS ONE

Journal Requirements:

Additional Editor Comments (if provided):

Reviewers' comments:

Reviewer's Responses to Questions

**Comments to the Author**

1. Does the manuscript provide a valid rationale for the proposed study, with clearly identified and justified research questions?

Reviewer #1: Partly

Reviewer #2: Yes

2. Is the protocol technically sound and planned in a manner that will lead to a meaningful outcome and allow testing the stated hypotheses?

Reviewer #1: Partly

Reviewer #2: Yes

3. Is the methodology feasible and described in sufficient detail to allow the work to be replicable?

Reviewer #1: Yes

Reviewer #2: Yes

4. Have the authors described where all data underlying the findings will be made available when the study is complete?

Reviewer #1: Yes

Reviewer #2: No

5. Is the manuscript presented in an intelligible fashion and written in standard English?

Reviewer #1: Yes

Reviewer #2: Yes

6. Review Comments to the Author

You may also provide optional suggestions and comments to authors that they might find helpful in planning their study.

Reviewer #1: The authors proposed a study to investigate how cerebellar inputs affect the motor cortical circuits producing indirect waves which are related to the cortical plasticity. The topic is interesting and the proposed study is worth doing.

1, My first major concern is that it is not clear whether this is a hypothesis driven or an exploratory protocol.

2, It seems that the central hypothesis is that cerebellar inputs change the production of later indirect wave. Related to this central hypothesis, I can not find a null hypothesis established by the authors. In particular, the last sentence of the introduction is difficult to understand.

3, There are a lot of measurements in the protocol. According to the null hypothesis, what is the primary outcome measure?

4, Sample size calculation is based on a previous study testing changes in short interval intracortical facilitation after anodal transcranial direct current stimulation. I do not understand the rationale behind this. Is this facilitation your primary outcome measure? May your study with cathodal stimulation have a different effect size from that with anodal stimulation?

Although this is a proposal and no data collection has been performed, some technical issues may be described in further details to ensure a publication.

5, Different stimulus intensities for the same cerebellar stimulation were described in the method. The authors may want to follow the detailed methods of previous studies (already cited in the manuscript) and define the intensity clearly. I agree with the opinion that the intensity should be tolerable. But low intensity stimulation will not produce cerebellar inhibition.

6, I do not understand how the sham stimulation may be performed in a way that no more than 2 repeats of any of 30 conditions are arranged in an intervention with 180 pairs of stimuli. With my calculation each condition will be repeated for 6 times in average.

7, I can not find where cerebellar direct current stimulation will be located.

8, Will the motor evoked potentials with posterior-anterior and anterior-posterior currents be measured in an active muscle? It may also need a little background introduction how such measurements are related to the later indirect wave recruitment.

9, What are the variables for regression analysis?

Reviewer #2: This is an interesting proposal and methodologically sound. I have listed a few recommendations / or points for consideration below. There may be no right answer to some of these suggestions.

I-wave excitability: “The stimulus intensity producing an MEP of approximately 0.5 mV in an active muscle will be assessed for both PA (MEPPA) an AP (MEPAP) orientations at baseline. A lower amplitude will be targeted in order to reduce stimulus intensity, and minimize concurrent recruitment of multiple I-waves.” I guess some researchers would consider contraction necessary while others might not. In my view, given the potential for contraction to interfere with the effects of some plasticity paradigms, I would personally avoid contraction pre- and post- ITMS/cTDCS. If it appears that equivalent results could be obtained and justified without contraction, then I would recommend this.

For the main SICF measures, one could consider aiming to reduce the number of pulses as in theory these could interact with the intervention.

ITMS ISI could be individualised for the 1.5 and 4.5ms ISI. It appears that a full ISI curve will be collected and SICF ISI individualised anyway. Some studies have found stronger effects, at least for ITMS1.5, when the ISI is individualised.

I think some previous ITMS studies used lower stimulus intensities, whereby e.g. “Stimulus intensity was the same for each pulse in the pair, and set so that paired stimulation generated a MEP of ~1 mV peak–peak amplitude.” The intensity required to achieve this may be less than that proposed here – I am not sure whether this is important. The authors could consider recording during ITMS and using this as a measure of SICF, provided a relevant baseline single pulse measure of equal intensity is available.

The sham condition could perhaps be designed such that it has an equal number of ISIs which fall at peaks or troughs. For example, if 2/3 of pulses happened to fall at “facilitatory” SICF intervals, it is conceivable that a somewhat “non-specific” facilitatory effect could occur. Also, I guess it is unclear whether this is relevant, but at ISIs >5ms, the MEP may begin to separate into two components – this may no longer involve what we typically think of as SICF.

Cathodal tDCS: In motor cortex, 2mA cathodal tDCS can elicit excitatory changes and differ to the depressive effects of 1mA tDCS. I am not sure whether this is the case for the Cerebellum, perhaps not.

With regard to post-intervention measures, the authors propose “Changes in SICF measures of I-wave excitability following the intervention will be investigated by assessing effects of iTMS session, time and ISI on baseline-normalised values.” I think this would be consistent with the study by Cash et al, 2009, where for the post-ITMS SICF curve “A similar analysis was performed on the post-intervention data, with the IPI data expressed as a percentage of the pre-intervention mean control value for each subject.” The authors of that paper explained this as follows “The correlation between the mean increase in the amplitude of the facilitation curve and the increase in the single-pulse MEP amplitude as a result of the intervention would mean that adjusting stimulus intensity based on the change in single pulse MEP amplitude would have obscured a change in the amplitude of the facilitation curves. Likewise, using the post-intervention single-pulse MEP as the control value for the facilitation curve post-intervention cancelled the increase in the I-wave peaks, consistent with the correlation between these measures.” I think the present authors are taking this point into account, but as it is a subtle point, I thought it might be worth flagging.

To demonstrate functional relevance, it could be interesting to include a behavioural measure or task. The authors mention relevance to motor learning. Perhaps a task could be performed early on to avoid interference with ITMS and again towards the end of the session to avoid and depotentiation etc. But this might miss the window of maximum plastic effects or increase the measurement of confounding fatigue during the course of the session. Also, perhaps difficult to find a task that works across repeated sessions.

7. PLOS authors have the option to publish the peer review history of their article (what does this mean?). If published, this will include your full peer review and any attached files.

Reviewer #1: No

Reviewer #2: Yes: Robin Cash

---

## [Author Response · Author response to Decision Letter 0]

26 May 2020

Summary

This document represent a point-by-point response to the concerns raised by two reviewers. As a result of these concerns, we have made substantial changes to the manuscript. In particular, clarification of the nature (exploratory vs hypothesis driven) and methodological approach of the proposed research has been provided for reviewer 1, whereas we have modified several aspects of the proposed methodology in response to the comments from reviewer 2. We thank the reviewers for their input and believe that these changes have substantially improved the quality of the manuscript.

Response to Reviewer 1

1. My first major concern is that it is not clear whether this is a hypothesis driven or an exploratory protocol.

Response:

We agree that this element of the project is poorly defined; although the submitted manuscript included specific hypotheses about the influence of the cerebellum on I-wave circuits, the project is intended to assess a number of mechanisms associated with this potential interaction. We therefore feel that it must be considered to be primarily exploratory in nature. Consequently, we have removed the specific hypotheses from the revised manuscript and reworded the final paragraph of the introduction to clarify the exploratory nature of the research. 

2. It seems that the central hypothesis is that cerebellar inputs change the production of later indirect wave. Related to this central hypothesis, I cannot find a null hypothesis established by the authors. In particular, the last sentence of the introduction is difficult to understand.

Response:

While we appreciate the point raised by the reviewer, the research is exploratory in nature and we therefore do not believe that inclusion of an alternative hypothesis is warranted.

3. There are a lot of measurements in the protocol. According to the null hypothesis, what is the primary outcome measure?

Response:

As the main aim of the proposed research is to assess if cerebellar modulation influences neuroplasticity of the I-wave generating circuits, specific measures of I-wave excitability are the main outcome of interest. In the context of our study, changes in SICF due to the iTMS intervention are therefore the primary outcome measure, and we have clarified this within the revised manuscript (page 6). However, given the exploratory nature of this research, this has not been framed in the context of a null hypothesis (see our response to the reviewer’s second concern). 

4. Sample size calculation is based on a previous study testing changes in short interval intracortical facilitation after anodal transcranial direct current stimulation. I do not understand the rationale behind this. Is this facilitation your primary outcome measure? May your study with cathodal stimulation have a different effect size from that with anodal stimulation?

Response:

As suggested by the reviewer, SICF is the primary outcome measure for the proposed study. This has been clarified in response to the reviewer’s third concern. Unfortunately, the exploratory nature of this research means that no previous work has investigated the influence of cerebellar tDCS (cathodal or anodal) on changes in SICF due to iTMS, and we are therefore unable able to base sample size estimates on an ideal dataset. However, the study by Ates et al. (2018) utilises cerebellar tDCS to modify the excitability of the I-wave generating circuits. We therefore believe that sample size calculations based on the effects of this study are sufficient to demonstrate the effects of activation within the pathway of interest (i.e., cerebellar projections to I-wave circuits of M1). While the Ates study does not inform sample sizes required to detect the response to iTMS, the proposed number of participants exceeds the numbers that have been included within previous studies reporting significant effects of this paradigm (more than doubling the sample in some instances). Consequently, the proposed sample size should be sufficient to detect effects of iTMS and effects of cerebellar modulation on SICF. We have clarified our use of the Ates study for sample size calculations within the revised manuscript (page 4).

5. Different stimulus intensities for the same cerebellar stimulation were described in the method. The authors may want to follow the detailed methods of previous studies (already cited in the manuscript) and define the intensity clearly. I agree with the opinion that the intensity should be tolerable. But low intensity stimulation will not produce cerebellar inhibition.

Response:

We agree with the reviewer’s suggestion, and have therefore modified the revised manuscript to state that a conditioning intensity of 60% MSO will be applied for measures of CBI. This intensity was chosen based on previous work characterising the recruitment of CBI, which showed that 60% MSO produces significant inhibition, but that intensities greater than this do not produce any more inhibition (Fernandez et al., 2018). Consequently, this level of stimulation finds a balance between achieving a significant level of CBI while maintaining tolerability for the participant. 

6. I do not understand how the sham stimulation may be performed in a way that no more than 2 repeats of any of 30 conditions are arranged in an intervention with 180 pairs of stimuli. With my calculation each condition will be repeated for 6 times in average.

Response:

Thanks to the reviewer for identifying this typographical error. As suggested, it should have read no more than 6 repeats. However, in response to reviewer 2’s fifth concern, an alternative approach to the sham stimulation has been adopted within the revised manuscript.

7. I cannot find where cerebellar direct current stimulation will be located.

Response:

Apologies to the reviewer for omitting this information. The cathode for cerebellar tDCS will be centred over the same location that is used for cerebellar TMS, 3 cm lateral and 1 cm inferior to the inion, on the line joining the inion and external auditory meatus of the right ear. This has been clarified within the revised manuscript (page 8)

8. Will the motor evoked potentials with posterior-anterior and anterior-posterior currents be measured in an active muscle? It may also need a little background introduction how such measurements are related to the later indirect wave recruitment.

Response:

The original protocol included PA/AP MEPs in an active muscle, as this approach minimises the need for multiple I-wave volleys to overcome the spinal motoneurone activation threshold, producing responses that are more likely to be reflective of inputs from single I-waves. However, in order to respond to reviewer 2’s first concern, these measures will now be recorded in a resting muscle. 

9. What are the variables for regression analysis?

Response:

Changes in CBI due to the intervention will be regressed against changes in measures of corticospinal (MEP1mV) and intracortical (SICF, MEPPA/AP) function in order to assess if alterations within the CB-M1 pathway contribute to plasticity effects. In addition, changes in intracortical function due to the intervention will be regressed against changes in corticospinal function in an attempt to identify if generalised changes in excitability are driven by changes in specific circuits. We have included this information in the revised manuscript (page 10/11). 

Also, post-intervention changes in the response to single and paired stimulation will be correlated against each other in order to select the most appropriate normalisation approach (see the response to reviewer 2’s seventh concern).

Response to Reviewer 2

1. I-wave excitability: “The stimulus intensity producing an MEP of approximately 0.5 mV in an active muscle will be assessed for both PA (MEPPA) an AP (MEPAP) orientations at baseline. A lower amplitude will be targeted in order to reduce stimulus intensity, and minimize concurrent recruitment of multiple I-waves.” I guess some researchers would consider contraction necessary while others might not. In my view, given the potential for contraction to interfere with the effects of some plasticity paradigms, I would personally avoid contraction pre- and post- ITMS/cTDCS. If it appears that equivalent results could be obtained and justified without contraction, then I would recommend this.

Response:

Thanks to the reviewer for their comment, it’s a very good point. Generally, we view muscle activation and low stimulus intensities as a necessity for ensuring measures that are I-wave selective when changing stimulus direction (Day et al., 1989). However, the main aim of the study is to assess the plasticity response, and the protocol must therefore be optimised to achieve this. We agree that muscle activation could interfere with any neuroplastic effects, and we will therefore apply directional measures in a resting muscle. Given the potential independence of the I-wave circuits activated with different current directions, we believe that these measures will still be of value for identifying contributions from different intracortical circuits. The revised manuscript has been updated to reflect these changes (page 7).

2. For the main SICF measures, one could consider aiming to reduce the number of pulses as in theory these could interact with the intervention.

Response:

We agree with the point raised by the reviewer and have therefore revised the protocol to include 12 trials per condition. However, given the variability of these measures, we would prefer not to reduce this number any further. 

3. ITMS ISI could be individualised for the 1.5 and 4.5ms ISI. It appears that a full ISI curve will be collected and SICF ISI individualised anyway. Some studies have found stronger effects, at least for ITMS1.5, when the ISI is individualised.

Response:

Apologies for the confusion, but we do not intend to record a full SICF curve. Only standard ISIs associated with the first (1.5 ms) and third (4.5 ms) peak will be tested. While we agree that a full curve would be interesting, and that individualised iTMS would be ideal, time limitations do not allow us to include these measures. If cerebellar modulation is able to influence iTMS effects, these are factors we intend to pursue further in future studies. 

4. I think some previous ITMS studies used lower stimulus intensities, whereby e.g. “Stimulus intensity was the same for each pulse in the pair, and set so that paired stimulation generated a MEP of ~1 mV peak–peak amplitude.” The intensity required to achieve this may be less than that proposed here – I am not sure whether this is important. The authors could consider recording during ITMS and using this as a measure of SICF, provided a relevant baseline single pulse measure of equal intensity is available.

Response:

Review of the iTMS literature indeed suggests that a large number of previous studies have employed lower intensity stimuli during the intervention. In line with Cash et al. (2009), we will therefore use paired stimuli producing an MEP of ~ 1 mV during the intervention. We intend to record responses during iTMS in order to track changes in excitability during the intervention. However, previous work suggests that single pulse stimulation applied at the intensity producing a response of ~1 mV as a pair does not produce an MEP (Thickbroom et al., 2006). It therefore seems likely that the MEP response to single-pulse stimulation applied at the intervention intensity will be too small for normalisation purposes. Consequently, we would prefer to retain the pre- and post-intervention SICF parameters as they were originally proposed. In contrast, changes in excitability during the intervention will be quantified by comparing the first, middle and last 12 stimuli. These alterations have been included within the revised manuscript (page 10).

5. The sham condition could perhaps be designed such that it has an equal number of ISIs which fall at peaks or troughs. For example, if 2/3 of pulses happened to fall at “facilitatory” SICF intervals, it is conceivable that a somewhat “non-specific” facilitatory effect could occur. Also, I guess it is unclear whether this is relevant, but at ISIs >5ms, the MEP may begin to separate into two components – this may no longer involve what we typically think of as SICF.

Response:

The reviewer raises a good point that we had not considered, and we have therefore modified the sham stimulation to counterbalance the number of facilitatory and inhibitory ISI’s. Facilitatory ISI’s of 1.5, 3 and 4.5 ms, and inhibitory ISI’s of 2 and 3.5 ms will be used during sham stimulation. These will be randomly applied, with 30 repeats for each facilitatory ISI and 45 repeats for each inhibitory ISI. These changes have been included in the revised manuscript (page 8).

6. Cathodal tDCS: In motor cortex, 2mA cathodal tDCS can elicit excitatory changes and differ to the depressive effects of 1mA tDCS. I am not sure whether this is the case for the Cerebellum, perhaps not.

Response:

Previous work suggests that disinhibitory effects of cathodal tDCS over cerebellum actually require a higher intensity of 2 mA (Galea et al., 2009; Fig 6A). While we agree that effects of tDCS on M1 have a complex relationship that is intensity-dependent, we therefore feel that this intensity is appropriate in this context. 

7. With regard to post-intervention measures, the authors propose “Changes in SICF measures of I-wave excitability following the intervention will be investigated by assessing effects of iTMS session, time and ISI on baseline-normalised values.” I think this would be consistent with the study by Cash et al, 2009, where for the post-ITMS SICF curve “A similar analysis was performed on the post-intervention data, with the IPI data expressed as a percentage of the pre-intervention mean control value for each subject.” The authors of that paper explained this as follows “The correlation between the mean increase in the amplitude of the facilitation curve and the increase in the single-pulse MEP amplitude as a result of the intervention would mean that adjusting stimulus intensity based on the change in single pulse MEP amplitude would have obscured a change in the amplitude of the facilitation curves. Likewise, using the post-intervention single-pulse MEP as the control value for the facilitation curve post-intervention cancelled the increase in the I-wave peaks, consistent with the correlation between these measures.” I think the present authors are taking this point into account, but as it is a subtle point, I thought it might be worth flagging.

Response:

Thanks to the reviewer for bringing this subtle point to our attention. We had originally planned to quantify paired-pulse facilitation relative to the single-pulse response recorded at the same time point, and to then normalise these values to baseline SICF. However, we will now assess the relationship between single and paired-pulse responses post-intervention, and quantify facilitation based on the pre-intervention single-pulse response if there is a significant correlation. If the correlation proves to be non-significant, we believe the original approach will be more appropriate. These changes have been included in the revised manuscript (page 9). 

8. To demonstrate functional relevance, it could be interesting to include a behavioural measure or task. The authors mention relevance to motor learning. Perhaps a task could be performed early on to avoid interference with ITMS and again towards the end of the session to avoid and depotentiation etc. But this might miss the window of maximum plastic effects or increase the measurement of confounding fatigue during the course of the session. Also, perhaps difficult to find a task that works across repeated sessions.

Response:

We agree that the inclusion of a behavioural measure would be very interesting. However, as suggested in response to the reviewer’s first concern, the main interest of this study is to assess the potential influence of cerebellum on plasticity mechanisms of M1, and we therefore need to optimise the protocol to achieve this. While it seems possible that distancing task performance in time may reduce its influence on the plasticity response, we cannot be sure to what extent this is true. Furthermore, as suggested by the reviewer, it is quite likely that this delay will reduce the likelihood of identifying any functional effects of the intervention. Finally, time constraints make it difficult to include additional measures without adding sessions, which we would prefer to avoid. Taken together, it seems that properly investigating the question of functional relevance will require a purpose-built experimental protocol. If the proposed intervention demonstrates physiological benefit, this is an avenue we will pursue further. 

References

Ates MP, Alaydin HC & Cengiz B. (2018). The effect of the anodal transcranial direct current stimulation over the cerebellum on the motor cortex excitability. Brain Res Bull 140, 114-119.

Cash R, Benwell N, Murray K, Mastaglia F & Thickbroom G. (2009). Neuromodulation by paired-pulse TMS at an I-wave interval facilitates multiple I-waves. Exp Brain Res 193, 1-7.

Day BL, Dressler D, Denoordhout AM, Marsden CD, Nakashima K, Rothwell JC & Thompson PD. (1989). Electric and magnetic stimulation of human motor cortex - surface EMG and single motor unit responses. J Physiol-London 412, 449-473.

Fernandez L, Major BP, Teo W-P, Byrne LK & Enticott PG. (2018). The impact of stimulation intensity and coil type on reliability and tolerability of cerebellar brain inhibition (CBI) via dual-coil TMS. Cerebellum 17, 540-549.

Galea JM, Jayaram G, Ajagbe L & Celnik P. (2009). Modulation of cerebellar excitability by polarity-specific noninvasive direct current stimulation. J Neurosci 29, 9115-9122.

Thickbroom GW, Byrnes ML, Edwards DJ & Mastaglia FL. (2006). Repetitive paired-pulse TMS at I-wave periodicity markedly increases corticospinal excitability: a new technique for modulating synaptic plasticity. Clin Neurophysiol 117, 61-66.

---

## [Decision Letter · Decision Letter 1]

17 Jun 2020

PONE-D-20-08985R1

Characterising the influence of cerebellum on the neuroplastic modulation of intracortical motor circuits.

PLOS ONE

Dear Dr. Opie,

Thank you for submitting your manuscript to PLOS ONE. After careful consideration, we feel that it has merit but does not fully meet PLOS ONE’s publication criteria as it currently stands. Therefore, we invite you to submit a revised version of the manuscript that addresses the points raised during the review process.

Reviewer 2 raised a point to be considered regarding sham stimulation.

We look forward to receiving your revised manuscript.

Kind regards,

Robert Chen

Academic Editor

PLOS ONE

Reviewers' comments:

Reviewer's Responses to Questions

**Comments to the Author**

1. Does the manuscript provide a valid rationale for the proposed study, with clearly identified and justified research questions?

Reviewer #1: Yes

Reviewer #2: Yes

2. Is the protocol technically sound and planned in a manner that will lead to a meaningful outcome and allow testing the stated hypotheses?

Reviewer #1: Yes

Reviewer #2: Yes

3. Is the methodology feasible and described in sufficient detail to allow the work to be replicable?

Reviewer #1: Yes

Reviewer #2: Yes

4. Have the authors described where all data underlying the findings will be made available when the study is complete?

Reviewer #1: Yes

Reviewer #2: Yes

5. Is the manuscript presented in an intelligible fashion and written in standard English?

Reviewer #1: Yes

Reviewer #2: Yes

6. Review Comments to the Author

You may also provide optional suggestions and comments to authors that they might find helpful in planning their study.

Reviewer #1: I do not have further comment.

I do not have further comment.

I do not have further comment.

I do not have further comment.

Reviewer #2: The protocol is well thought out and the authors have done a great and timely job of responding.

I have one minor comment with regard to sham stimulation - depending on the intensities used for SICF/ITMS, paired pulse ISIs could be facilitatory, rather than inhibitory. I think this point might be illustrated in the figures of Ilic and colleagues (2002). E.g. I think it is even possible to have facilitation at ISI 2ms, depending on the intensity. Personally I would probably avoid including ISI 1.5ms (or any in the range of 1.1-1.7ms) in the sham condition and try to select ISIs that are intermediate to peaks and troughs - ISIs that would be anticipated to show little facilitation or inhibition at the intensities employed. Ultimately, there is probably no ideal way to circumvent this issue but I raise it for the author's consideration.

7. PLOS authors have the option to publish the peer review history of their article (what does this mean?). If published, this will include your full peer review and any attached files.

Reviewer #1: No

Reviewer #2: Yes: Robin Cash

---

## [Author Response · Author response to Decision Letter 1]

18 Jun 2020

Reviewer 2 comment:

I have one minor comment with regard to sham stimulation - depending on the intensities used for SICF/ITMS, paired pulse ISIs could be facilitatory, rather than inhibitory. I think this point might be illustrated in the figures of Ilic and colleagues (2002). E.g. I think it is even possible to have facilitation at ISI 2ms, depending on the intensity. Personally I would probably avoid including ISI 1.5ms (or any in the range of 1.1-1.7ms) in the sham condition and try to select ISIs that are intermediate to peaks and troughs - ISIs that would be anticipated to show little facilitation or inhibition at the intensities employed. Ultimately, there is probably no ideal way to circumvent this issue but I raise it for the author's consideration.

Response:

Thanks to the reviewer for their additional comment. As suggested, the issue of sham stimulation for the iTMS protocol represents a significant challenge. However, we appreciate the point being made, and will therefore modify the sham protocol to only use ISI’s that represents transitions between peaks and troughs of facilitation. Consequently, the sham protocol will include equal repetitions of ISI’s 1.8, 2.3, 3.3, 3.8 and 4.7 ms. These intervals were identified based on facilitation curves presented by several previous studies (Ziemann et al., 1998; Delvendahl et al., 2014; Cirillo et al., 2015; Opie et al., 2018). The revised manuscript has been updated to include these details (page 8). 

Cirillo J, Calabro FJ & Perez MA. (2015). Impaired organization of paired-pulse TMS-induced I-waves after human spinal cord injury. Cereb Cortex 26, 2167-2177.

Delvendahl I, Lindemann H, Jung NH, Pechmann A, Siebner HR & Mall V. (2014). Influence of waveform and current direction on short-interval intracortical facilitation: a paired-pulse TMS study. Brain Stimul 7, 49-58.

Opie GM, Cirillo J & Semmler JG. (2018). Age‐related changes in late I‐waves influence motor cortex plasticity induction in older adults. J Physiol 596, 2597-2609.

Ziemann U, Tergau F, Wassermann EM, Wischer S, Hildebrandt J & Paulus W. (1998). Demonstration of facilitatory I wave interaction in the human motor cortex by paired transcranial magnetic stimulation. J Physiol 511, 181-190.

---

## [Editor Report · Decision Letter 2]

29 Jun 2020

Characterising the influence of cerebellum on the neuroplastic modulation of intracortical motor circuits.

PONE-D-20-08985R2

Dear Dr. Opie,

We’re pleased to inform you that your manuscript has been judged scientifically suitable for publication and will be formally accepted for publication once it meets all outstanding technical requirements.

Kind regards,

Robert Chen

Academic Editor

PLOS ONE

---

## [Editor Report · Acceptance letter]

1 Jul 2020

PONE-D-20-08985R2 

Characterising the influence of cerebellum on the neuroplastic modulation of intracortical motor circuits. 

Dear Dr. Opie:

I'm pleased to inform you that your manuscript has been deemed suitable for publication in PLOS ONE. Congratulations! Your manuscript is now with our production department. 

Kind regards, 

on behalf of

Dr. Robert Chen 

Academic Editor

PLOS ONE